# Colorimetric Chemosensor Based on Fe_3_O_4_ Magnetic Molecularly Imprinted Nanoparticles for Highly Selective and Sensitive Detection of Norfloxacin in Milk

**DOI:** 10.3390/foods12020285

**Published:** 2023-01-07

**Authors:** Maiquan Li, Lingli Luo, Jiayin Li, Yingzi Xiong, Ling Wang, Xia Liu

**Affiliations:** Hunan Provincial Key Laboratory of Food Science and Biotechnology, College of Food Science and Technology, Hunan Agricultural University, Changsha 410128, China

**Keywords:** Fe_3_O_4_ MMIP NPs, peroxidase-like catalytic activity, colorimetric chemosensor, norfloxacin detection

## Abstract

Long-term use of norfloxacin (NOR) will cause NOR residues in foods and harm human bodies. The determination of NOR residues is important for guaranteeing food safety. In this study, a simple, selective, and label-free colorimetric chemosensor for in situ NOR detection was developed based on Fe_3_O_4_ magnetic molecularly imprinted nanoparticles (Fe_3_O_4_ MMIP NPs). The Fe_3_O_4_ MMIP NPs showed good peroxidase-like catalytic activity to 3,3′,5,5′-tetramethylbenzidine (TMB) and selective adsorption ability to NOR. The colorimetric chemosensor was constructed based on the Fe_3_O_4_ MMIP NPs-H_2_O_2_-TMB reaction system. The absorbance differences were proportional to the concentrations of NOR in the range of 10–300 ng/mL with a limit of detection at 9 ng/mL. The colorimetric chemosensor was successfully applied to detect NOR residue in milk. The recovery range was 78.2–95.81%, with a relative standard deviation of 2.1–9.88%. Together, the proposed colorimetric chemosensor provides a reliable strategy for the detection of NOR residues in foods.

## 1. Introduction

Norfloxacin (NOR) was the first discovered fluoroquinolone antibiotic [1] and is widely used in the treatment of human and animal respiratory, digestive, and urinary tract systems [2]. Long-term use of NOR can easily induce bacterial resistance, allergic reactions, or serious harm, such as the “three-cause” effect (carcinogenic, teratogenic, and mutagenic) on the human body. Notably, NOR will increase human resistance to drugs and induce clinical treatment difficulties, which are serious threats to public health. In 2005, the European Union regulated that the maximum residue level of NOR in animal products must not exceed 0.1 ppm [3]. In 2015, the Ministry of Agriculture of the People’s Republic of China issued regulations on the suspension of production, operations, and usage of NOR in animal products in announcement no. 2292 [4]. However, food safety problems caused by the residuals of NOR in animal-derived foods still exist. Therefore, the determination of NOR residue has become an important task in food safety.

The reported detection methods of NOR, such as chromatography [5,6], biosensors [7], and an enzyme-linked immunosorbent assay (ELISA) [8] are usually expensive and time-consuming; moreover, the sample pretreatment is complicated. Furthermore, the practical applications of the biosensor and ELISA methods are severely restricted because the natural enzymes they need are usually too complex to prepare and purify, poor in stability, and expensive. To overcome these shortcomings, researchers have found a series of mimetic enzymes that show similar catalytic activity with natural enzymes [9] and even higher stability than natural enzymes, such as cyclodextrin, crown ethers, and porphyrins. Nevertheless, there are still some problems to be solved, such as the low catalytic efficiency and complex synthesis process.

The new generation of mimetic enzymes (nanozymes) [10,11,12,13] can overcome some shortcomings of mimetic enzymes. At present, more nanomaterials with peroxidase-like catalytic activities are used as nanozymes, such as vanadium pentoxide [14], copper oxide [15], manganese dioxide [16], cobalt tetroxide [17,18], cerium oxide [19], and precious metals, such as platinum [20], gold [21], etc. Among them, Fe_3_O_4_ magnetic nanoparticles (Fe_3_O_4_ NPs) are favored by researchers due to their dual catalytic properties of both peroxidase (POD) and catalase (CAT), as well as good stability and superparamagnetism.

In recent years, the colorimetric method based on Fe_3_O_4_ NPs has attracted much attention. For example, the colorimetric method based on Fe_3_O_4_ NPs showed high sensitivity and selectivity toward glutathione [22]. The colorimetric method for the detection of *S. typhimurium*, based on Fe_3_O_4_ NPs and DNA aptamers, is cost-effective, simple, and able to visibly detect bacteria up to 7.5 × 10^5^ CFU/mL [23]. The colorimetric method based on vitamin C (V_C_)-functionalized Fe_3_O_4_ NPs (VcFe_3_O_4_ NPs) exhibits good sensitivity in detecting H_2_O_2_ and glucose as low as 0.29 and 0.288 μmol/L, respectively [24]. However, Fe_3_O_4_ NPs are still restricted to specific analytes, require coupled antibodies, or need complex sample pretreatments.

Magnetic molecularly imprinted polymer nanoparticles (MMIP NPs) are developed via MNPs combined with molecularly imprinted polymers (MIPs), which show peroxidase-like catalytic activity, specific adsorption, and magnetic separation ability. MMIP NPs have been successfully applied in the colorimetric sensor. Kong et al. [25] constructed a paper-based colorimetric sensor for the detection of bisphenol A (BPA) based on the peroxidase-like catalytic activity of ZnFe_2_O_4_ and cellulose paper wrapped with MIPs. Fan et al. [26] fabricated MIP PtPd nanoflowers with peroxidase-like catalytic activity for selective recognition and detection of H_2_O_2_ and glucose. Guo et al. [27] developed a selective analytical method for puerarin by enwrapping a PtCu/PSS-Gr nanocomposite with MIPs and combining it with the high peroxidase-like catalytic activity of PtCu/PSS-Gr. Therefore, MMIP NPs will provide new possibilities in the application of Fe_3_O_4_ NPs.

In this work, a new colorimetric chemosensor for NOR detection was constructed based on Fe_3_O_4_ MMIP NPs. Fe_3_O_4_ MMIP NPs were synthesized based on Fe_3_O_4_ NPs as magnetic carriers by surface imprinting technology, and their peroxidase-like catalytic activities were examined. The experimental conditions of detecting NOR were optimized. The detection performance of the colorimetric chemosensor was explored. Moreover, the applicability of the colorimetric chemosensor was demonstrated by detecting NOR in milk and compared with the ELISA method. The contrasted colorimetric chemosensor method proved to be simple, fast, low-cost, and efficient for NOR detection in foods.

## 2. Materials and Methods

### 2.1. Reagents and Instruments

Norfloxacin (NOR), ciprofloxacin (CIP), enrofloxacin (ENR), and danofloxacin (DAN) were purchased from Shanghai Yuanye Bio-Chem Technology Co., Ltd. (Shanghai, China) and the purity was >98%. Tetramethyl ammonium hydroxide (TMAOH) and (3-Aminopropyl) triethoxysilane (APTES) were purchased from Shanghai Aladdin Bio-Chem Technology Co., Ltd. (Shanghai, China). Dopamine (DA), tetracycline hydrochloride (TC), and sulfadiazine (SD) were purchased from Shanghai Aladdin Bio-Chem Technology Co., Ltd. (Shanghai, China), and the purity was >98%. TMB, OPD, and ABTS were purchased from Shanghai Macklin Bio-Chem Co., Ltd. (Shanghai, China) and the purity was >99%. A NOR ELISA Kit (RN 45S) was purchased from Shenzhen Rongjin Technology Co., Ltd. All chemicals and solvents were of a commercially available analytical reagent grade; double distilled water was used throughout this work.

The milk samples were natural pure milk purchased from Inner Mongolia Mengniu Dairy (Group) Limited by Share, Ltd., natural pure milk purchased from Inner Mongolia Yili Industrial Group Limited by Share, Ltd., and deluxe milk purchased from Inner Mongolia Mengniu Dairy (Group) Limited by Share, Ltd.

The absorbance spectrum was obtained by a multi-mode microplate spectrophotometer (Multiskan GO 1510, Thermo Fisher Scientific, Waltham, MA, USA).

### 2.2. Synthesis of Fe_3_O_4_ MMIP NPs

Fe_3_O_4_ MMIP NPs were prepared according to the procedure that our group previously established [28]. Briefly, FeCl_3_∙6H_2_O and FeCl_2_∙4H_2_O were mixed and heated under nitrogen gas protection. Then, 40 mL of NaOH (2 mol/L) was added, and this mixture was stirred (550 rpm, 1 h, 80 °C). Fe_3_O_4_ NPs were dissolved in TMAOH solution (7%), followed by 2 h of incubation; they were then dissolved in a 120 mL solution of ethanol–water (1:1, *v*/*v*) (containing 100 mM Tris-HCl, pH = 8.5). Subsequently, 3 mL of APTES was added and this mixture was stirred (40 °C, 350 rpm, 10 h). A total of 500 mg of APTES-modified Fe_3_O_4_ NP was dissolved in 100 mL of ethanol–water (1:1, pH = 8.5). After ultrasonic treatment (25 °C,10 min), 15 mg of NOR and 135 mg of DA were added to the flask. The mixture was stirred (350 rpm, 3 h, 25 °C). Then, the Fe_3_O_4_ MMIP NPs were separated, dried in a vacuum (60 °C, 24 h), and stored at 4 °C. Meanwhile, the Fe_3_O_4_ magnetic non-imprinting polymer nanoparticles (Fe_3_O_4_ MMIP NPs) were prepared without the template molecule as a control.

### 2.3. Peroxidase-Like Catalytic Activity of Fe_3_O_4_ MMIP NPs

Chromogenic agents were examined through the Fe_3_O_4_ MMIP NP catalytic oxidation experiment: 200 µL of Fe_3_O_4_ MMIP NPs (4 mg/mL), 100 µL of ultrapure water, 400 µL of HAc-NaAc (0.2 M), 250 µL of H_2_O_2_ (0.06 M), and 50 µL of a substrate (0.008 M TMB, OPD or ABTS) was mixed and incubated for 4 min at 25 °C. The supernatant was separated, then its absorbance spectrum was measured by a multi-mode microplate spectrophotometer. 

The feasibility of the Fe_3_O_4_ MMIP NPs for NOR detection was verified by investigating the absorbance difference (ΔA) of the Fe_3_O_4_ MMIP NP-H_2_O_2_-TMB reaction system before and after the addition of NOR. A total of 100 µL of NOR (0.01 mg/mL) or ultrapure water was added to 200 µL of Fe_3_O_4_ MMIP NPs/Fe_3_O_4_ MMIP NPs (4 mg/mL) and stirred for 15 min in a constant temperature incubator. The supernatant was separated, then its absorbance spectrum was measured via a multi-mode microplate spectrophotometer. 

### 2.4. In Situ Colorimetric Measurement

The effects of the buffer type (NaAc, HAc-NaAc, Na_2_HPO_4_-CA, or Na_2_HPO_4_-NaH_2_PO_4_), pH (2–6), H_2_O_2_ concentration (0.004–2 M), the volume ratio of TMB (0.008 M) to H_2_O_2_ (0.1 M) (1:1, 1:2, 1:3, 1:4, 4:1, 3:1, and 2:1), and reaction time (1, 2, 3, 4, 6, 8, 10, 12, and 14 min) on the catalytic activity of Fe_3_O_4_ MMIP NPs were investigated. Under the optimal catalytic activity conditions, the colorimetric detection of NOR based on the Fe_3_O_4_ MMIP NPs was achieved. Moreover, 10 µL of a standard solution containing different concentrations of NOR was separately incubated with 200 µL of Fe_3_O_4_ MMIP NPs. Then, 200 µL of HAc-NaAc, 400 µL of H_2_O_2_, and 100 µL of TMB were added. The mixture was incubated for 4 min at 25 °C. The supernatant was then separated and the absorbance intensity at 652 nm was measured by a multi-mode microplate spectrophotometer. The selectivity experiments were carried out using 100 µL of a standard solution of NOR, CIP, ENR, DAN, SD, and TC, with an initial concentration of 200 ng/mL. Reproducibility experiments were also carried out in the same procedure. In the reproducibility experiments, the Fe_3_O_4_ MMIP NPs were separated by a magnet and washed twice with 1 mL of ultrapure water.

### 2.5. Preparation of Milk Samples

The milk samples were purchased from a local supermarket. A total of 1 mL of milk was vortically mixed with 8 mL of an EDTA-McIlvaine buffer solution (1000 r/min, 1 min), followed by an ultrasonic for 10 min (25 °C, 60 Hz), and centrifugated for 10 min (10,000 r/min, 4 °C). Then the supernatant with different concentrations of NOR (10, 50, and 100 ng/mL) was prepared for the recovery experiments. The absence of NOR in the milk sample was confirmed by high-performance liquid chromatography-tandem mass spectrometry (GB/T21312-2007, China).

### 2.6. Calculation of LOD

According to the rules of the International Union of Pure and Applied Chemistry (IUPAC): *C_L_ = k × S_b_/m* (*C_L_* means the detection limit; k means the confidence factor; *m* means the slope of the standard curve in the low concentration range; *S_b_* means the standard deviation of a blank).

## 3. Results and Discussion

### 3.1. The Kinetic Parameters of TMB Oxidation Catalyzed by Fe_3_O_4_ MMIP NPs

In order to explain the peroxidase-like catalytic activity of Fe_3_O_4_ MMIP NPs and construct a colorimetric chemosensor, the kinetic parameters of TMB oxidation catalyzed by Fe_3_O_4_ MMIP NPs were determined. The initial reaction velocity was calculated according to the Michaelis–Menten equation: v = Vmax [S]/(Km + [S]) (v is the initial reaction velocity, Vmax is the maximum reaction rate, [S] is the substrate concentration, Km is the Michaelis constant.) The relationship between the H_2_O_2_ concentration (or the TMB concentration) and the reaction rate was obtained using the double-reciprocal mapping method.

Michaelis–Menten curves were obtained (Appendix A). Km and Vmax were obtained by the Lineweaver–Burk plot (Table 1). The Km of Fe_3_O_4_ MMIP NPs to TMB and H_2_O_2_ (0.088 mM and 0.16 mM, respectively) was smaller than that of HRP [28], indicating that the affinity of Fe_3_O_4_ MMIP NPs for TMB and H_2_O_2_ was higher than that of HRP, and Fe_3_O_4_ MMIP NPs could catalyze the oxidation of TMB faster than HRP. Compared with other Fe-based nanomaterials [24,29,30,31,32], the Fe_3_O_4_ MMIP NPs in this work also showed relatively high affinity and a maximum reaction rate to TMB and H_2_O_2._ The reason may be that small particle sizes, large selective surface areas, and rich surface charges were easily able to adsorb positively charged TMB and had a high affinity for TMB [33]. 

### 3.2. Detection Feasibility

TMB, o-phenylenediamine (OPD), and 2,2′-diazo-bis-3-ethylbenzothiazoline-6-sulfonic acid (ABTS) were used to verify the peroxide-like catalytic activity of Fe_3_O_4_ MMIP NPs (Figure 1A). In the participation of H_2_O_2_, Fe_3_O_4_ MMIP NPs could catalyze the oxidation of TMB, OPD, and ABTS, which made the solution appear blue, yellow, and green, respectively. The above results indicate that Fe_3_O_4_ MMIP NPs has peroxidase-like catalytic activity. The absorbance of the Fe_3_O_4_ MMIP NPs-H_2_O_2_-TMB reaction system at the optimal absorption wavelength (652 nm) is much more obvious than that of the Fe_3_O_4_ MMIP NPs-H_2_O_2_-OPD reaction system (450 nm) and Fe_3_O_4_ MMIP NPs-H_2_O_2_-ABTS reaction system (415 nm or 734 nm). Therefore, TMB was selected as the chromogenic agent for Fe_3_O_4_ MMIP NPs. 

Then, the feasibility of the NOR detection of the constructed colorimetric chemosensor was verified by catalyzing the oxidation of colorless TMB into a blue product (oxTMB). As shown in Figure 1B, when there were only Fe_3_O_4_ MMIP NPs or H_2_O_2_ in the reaction system, no absorbance at 652 nm could be observed and the solution was colorless, which indicated that TMB can only be oxidized when Fe_3_O_4_ MMIP NPs and H_2_O_2_ exist simultaneously (curve e, curve f, and curve g). Compared with curve g, curve h showed no absorbance at 652 nm; this might be caused by the direct inhibiting effect of NOR on the TMB-H_2_O_2_ reaction. On the contrary, when Fe_3_O_4_ MMIP NPs, TMB, and H_2_O_2_ existed, the absorbance of the reaction system at 652 nm was greatly enhanced (curve a), which indicated that Fe_3_O_4_ MMIP NPs had high peroxidase-like catalytic activity to catalyze the oxidation of TMB in the participation of H_2_O_2_. When NOR was added to the reaction system, the absorbance at 652 nm was reduced (curve b) and the solution color became lighter, indicating that the presence of NOR inhibited the peroxidase-like catalytic activity of Fe_3_O_4_ MMIP NPs. The reason was that NOR could selectively bind to the cavity in the surface of Fe_3_O_4_ MMIP NPs, which reduced the contact area of the H_2_O_2_ and Fe_3_O_4_ MMIP NPs. As a comparison, the peroxidase-like catalytic activity of Fe_3_O_4_ MMIP NPs was also tested. Compared with curve c, the absorbance peak intensity of curve d was lower. This may have been caused by unselective binding between NOR and Fe_3_O_4_ MMIP NPs. It is worth noting that the absorption peak intensities of curve c and curve d were lower than those of curve a and curve b, indicating that the peroxidase-like catalytic activities of Fe_3_O_4_ MMIP NPs were lower than those of Fe_3_O_4_ MMIP NPs. Due to the lack of selective cavities on the surfaces of Fe_3_O_4_ MMIP NPs, a barrier was formed between H_2_O_2_ and Fe_3_O_4_ MMIP NPs, which hindered the catalysis of Fe_3_O_4_ MMIP NPs to the substrate. All of the above results indicate that the Fe_3_O_4_ MMIP NPs-H_2_O_2_-TMB reaction system has feasibility for the colorimetric detection of NOR.

### 3.3. Optimization of the Experimental Conditions

Buffer types affected the peroxidase-like catalytic activities of Fe_3_O_4_ MMIP NPs greatly (Figure 2A). The highest absorbance was observed in HAc-NaAc (0.2 M). We speculated that the concentration change of H^+^ promoted the dissociation of H_2_O_2_, which accelerated the oxidation rate of TMB. Therefore, HAc-NaAc (0.2 M) was selected as the buffer for the reaction system. 

The effect of pH on the peroxidase-like catalytic activity of Fe_3_O_4_ MMIP NPs was examined by the buffer within the pH range of 2–6. Figure 2B shows the pH-dependent response curve. When pH < 3, the absorbance of the reaction system increased, accompanied by the increase of the pH, while the further increasing pH resulted in a decrease of the absorbance intensity. Therefore, pH = 3 was selected as the condition for the reaction system.

The TMB/H_2_O_2_ volume ratio also had a great effect on the peroxidase-like catalytic activity of Fe_3_O_4_ MMIP NPs (Figure 2C). The absorbance of the reaction system increased with the increase of the H_2_O_2_ volume, and the absorbance reached the maximum when the volume ratio of TMB/H_2_O_2_ was 1:4. When the volume of H_2_O_2_ was fixed, the volume of TMB increased, and the absorbance value showed a decreasing trend. Therefore, the volume ratio of TMB to H_2_O_2_ was chosen as 1:4 for the reaction system. 

The influence of the H_2_O_2_ concentration on the peroxidase-like catalytic activity of Fe_3_O_4_ MMIP NPs was studied and the results are shown in Figure 2D. With the increase of the H_2_O_2_ concentration (<0.1 M), the absorbance increased rapidly. When the concentration of H_2_O_2_ was 0.1 M, the absorbance value was 0.7943 and the solution color changed significantly in a short time, which was easy to be recognized by the naked eye. When the concentration was more than 0.1 M, the increase rate decreased and gradually leveled off, and the color change with the concentration of H_2_O_2_ was not obvious. Moreover, 0.1 M H_2_O_2_ was chosen for the reaction system. 

The reaction time of the Fe_3_O_4_ MMIP NPs-H_2_O_2_-TMB system was examined from 1 to 14 min. The absorbance increased gradually when the reaction time was prolonged (Figure 2E). When the reaction time was 4 min, the solution color changed sufficiently (it could be recognized by the naked eye). Considering the purpose of rapid detection, 4 min was chosen as the reaction time for the reaction system.

### 3.4. In Situ Colorimetric Measurements

Under the optimized experimental conditions, different concentrations of NOR were detected by the constructed colorimetric chemosensor based on Fe_3_O_4_ MMIP NPs. As can be seen in Figure 3A, with the increase in the NOR concentration, the absorbance of the reaction system at 652 nm gradually decreased. This indicated that NOR inhibited the production of oxTMB. There are two main reasons. First, NOR occupied the cavities on the surface of Fe_3_O_4_ MMIP NPs and prevented the contact between H_2_O_2_ and Fe3O4 MMIP NPs, which partially decreased the peroxide-like catalytic activity of Fe_3_O_4_ MMIP NPs. Second, there was a direct inhibiting effect of NOR on the TMB-H_2_O_2_ reaction. Figure 3B shows a good linear relationship between the ΔA (652 nm) and NOR concentration in the range of 10–300 ng/mL. The linear regression equation was Y = 0.000947x + 0.03176 (R^2^ = 0.9878), and the detection limit was 9 ng/mL.

To evaluate the selectivity of the colorimetric chemosensor, the structural analogs (CIP, ENR, and DAN) and nonstructural analogs (SD and TC) were selected as interfering antibiotics. Figure 3C shows the ΔA of the Fe_3_O_4_ MMIP NPs-H_2_O_2_-TMB reaction system with and without NOR, CIP, ENR, DAN, SD, and TC (the final concentration of each component was 200 ng/mL). The ΔA of the colorimetric chemosensor for NOR detection was the largest (1.77 times, 2.27 times, 2.67 times, 6.41 times, and 11.42 times for CIP, ENR, DAN, SD, and TC, respectively). This is because Fe_3_O_4_ MMIP NPs could selectively adsorb NOR. The results clearly demonstrate the selectivity of the Fe_3_O_4_ MMIP NP-based colorimetric chemosensor. In addition, the ΔA of the Fe_3_O_4_ MMIP NPs-H_2_O_2_-TMB reaction system was less than that of the Fe_3_O_4_ MMIP NPs-H_2_O_2_-TMB system under the same conditions. This was mainly because the Fe_3_O_4_ MMIP NPs failed to form molecular imprint cavities, and there was only a small amount of non-selective adsorption.

The reusability assay (Figure 3D) showed that the catalytic activities of Fe_3_O_4_ MMIP NPs remained above 86% after 6 cycles of colorimetric detection, indicating that the Fe_3_O_4_ MMIP NP-based colorimetric chemosensor had good stability. Xiong et al. [34] revealed that no loss of activity was observed after 10 cycles of the magnetic core–shell nanoflower Fe_3_O_4_@MnO_2._ The difference might have been induced by the synthesis method. However, Lian et al. [35] reported that when the relative activity was more than 80%, the suitable stability was good.

### 3.5. Detection of NOR in Milk

After the pretreatment of three different milk samples, the constructed colorimetric chemosensor was applied to detect the NOR residuals in the samples. The results (shown in Appendix A) show that no response signal (ΔA) was obtained, indicating that there were no NOR residues in the samples, which was consistent with the results of the ELISA kit (the detected NOR concentrations in three milk samples were −0.0906 ng/mL, −0.1660 ng/mL, and −0.1376 ng/mL, respectively).

The recovery experiments showed that the recovery range of the established colorimetric method was 78.29% to 95.81%, and the relative standard deviation (RSD) ranged from 3.47% to 9.88% (Table 2). The constructed colorimetric chemosensor had good reliability and the detection time was short (only 30 min). The results above prove that this colorimetric chemosensor can be used for the rapid determination of NOR residue in milk.

### 3.6. Comparison of Detection Methods for NOR

The colorimetric chemosensor was compared with other methods for NOR detection in animal-derived foods (Table 3). While HPLC analysis [36], HPLC with fluorescence [37], ELISA method [38], the fluorescence analysis [8,39,40], and Raman spectroscopy [41] improved the sensitivity and accuracy for NOR detection, they increased the testing costs as well as the requirements for operators. Moreover, the antibodies used in the ELISA method increased the testing costs, and the probe based on quantum dots used in the fluorescence analysis made the procedure much more complex. Xie et al. [42] proposed a rapid method for the preparation of novel MIPs with covalent organic frameworks as support for the selective recognition of NOR. However, the synthesis of MIPs with covalent organic frameworks is complex, the separation of MIPs from the samples needs centrifugation, and the quantitative analysis needs an HPLC analysis.

Compared with the studies above, the colorimetric chemosensor in this work showed several obvious advantages. First, the separation of Fe_3_O_4_ MMIP NPs from samples by magnets was quick. Second, the Fe_3_O_4_ MMIP NPs could selectively absorb NOR from samples and had good stability. Third, the sample pretreatment was perfectly combined with colorimetric detection making the whole procedure simple. Finally, the visual qualitative analysis could be achieved by the color change while quantitative analysis could be realized via simple instruments. Therefore, the colorimetric chemosensor could offer a better way for NOR detection in food.

## 4. Conclusions

In summary, a new colorimetric chemosensor for in situ detection of NOR based on the peroxidase-like catalytic activities of Fe_3_O_4_ MMIP NPs was constructed. Fe_3_O_4_ MMIP NPs have highly selective absorption abilities, excellent catalytic performances, and good stability for NOR detection. The detection procedure and the sample pretreatment are simple. This method has been successfully applied for the selective recognition and determination of NOR residue in milk and the result is consistent with the ELISA kit. The established method can provide a reliable strategy for the simple, fast, and precise detection of NOR residue in milk.

## Figures and Tables

**Figure 1 foods-12-00285-f001:**
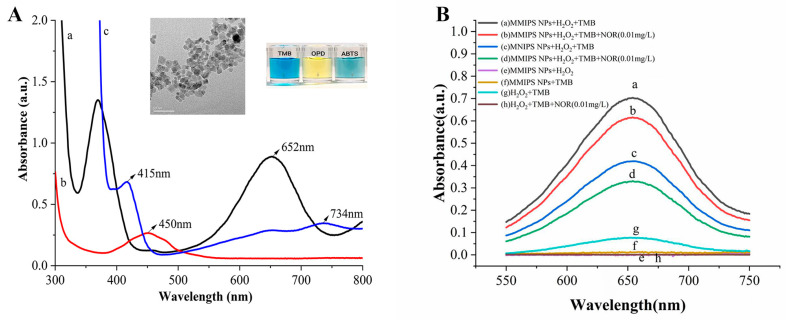
Detection feasibility of Fe_3_O_4_ MMIP NPs for NOR. (**A**) Absorption spectra of the reactions of different chromogenic substrates, a: TMB, b: OPD, and c: ABTS. (**B**) Absorption spectra of different reaction systems a: Fe_3_O_4_ MMIP NPs + H_2_O_2_ + TMB, b: Fe_3_O_4_ MMIP NPs + H_2_O_2_ + TMB + 0.01 mg/mL NOR, c: Fe_3_O_4_ MMIP NPs + H_2_O_2_ + TMB, d: Fe_3_O_4_ MMIP NPs + H_2_O_2_ + TMB + 0.01 mg/mL NOR, e: Fe_3_O_4_ MMIP NPs + H_2_O_2_, f: Fe_3_O_4_ MMIP NPs + TMB, g: H_2_O_2_ + TMB, h: H_2_O_2_ + TMB + 0.01 mg/mL NOR. All trials were repeated 3 times.

**Figure 2 foods-12-00285-f002:**
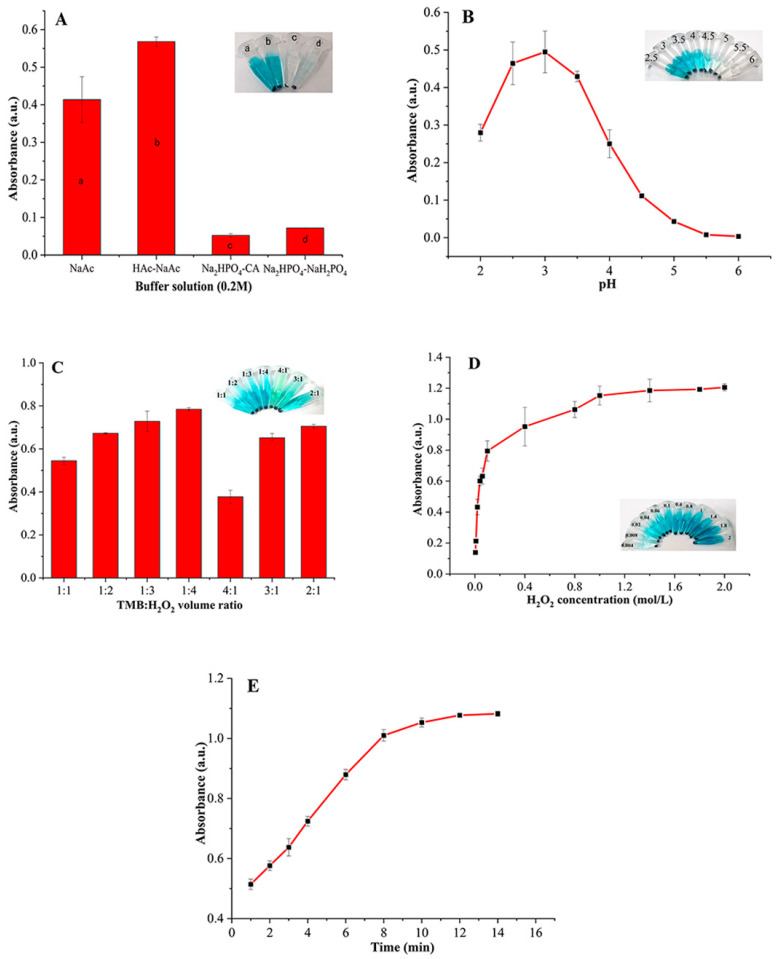
The influences of different factors on the performances of Fe_3_O_4_ MMIP NPs. (**A**) Buffer type, 0.2 M buffer pH = 3.5, 250 µL 0.06 M H_2_O_2_, 50 µL 0.008 M TMB, reaction time 4 min; (**B**) pH, 0.2 M HAc-NaAc, 250 µL 0.06 M H_2_O_2_, 50 µL 0.008 M TMB, reaction time 4 min; (**C**) TMB/H_2_O_2_ volume ratio, 0.2 M HAc-NaAc pH = 3.5, 0.06 M H_2_O_2_, 0.008 M TMB, reaction time 4 min; (**D**) H_2_O_2_ concentration, 0.2 M HAc-NaAc pH = 3.5, 250 µL H_2_O_2_, 50 µL 0.008 M TMB, reaction time 4 min; (**E**) Reaction time, 0.2 M HAc-NaAc pH = 3.5, 250 µL 0.06 M H_2_O_2_, 50 µL 0.008 M TMB. All trials were repeated 3 times.

**Figure 3 foods-12-00285-f003:**
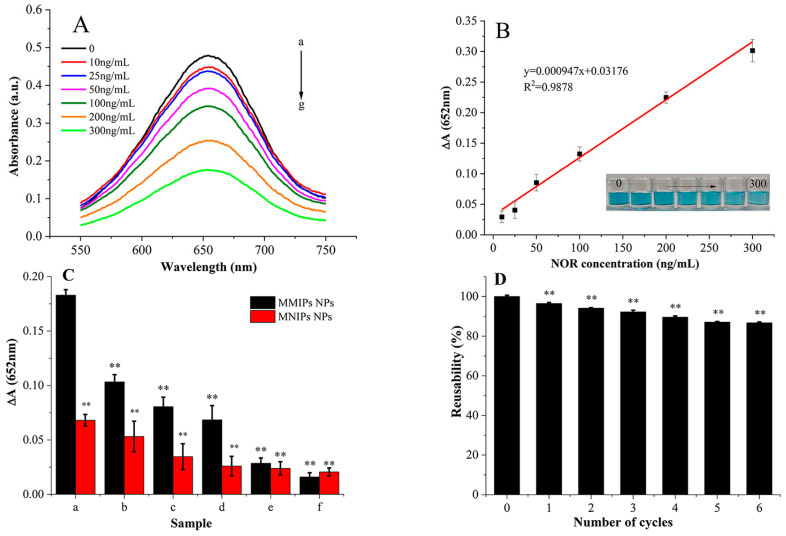
In situ colorimetric measurements. (**A**) Absorption spectra of oxTMB at different NOR concentrations (a–g: 0, 10, 25, 50, 100, 200, and 300 ng/mL). (**B**) Linear relationship between the ΔA (652 nm) and NOR concentrations. (**C**) The selectivity of the colorimetric chemosensor at 25 °C (a: NOR, b: CIP, c: ENR, d: DAN, e: SD, f: TC, with the initial concentration of 0.02 mg/mL), ** significantly different compared with MMIP NPs to NOR, *p* < 0.01. (**D**) The reusability of the Fe_3_O_4_ MMIP NPs for NOR, ** significantly different compared to the first time used, *p* < 0.01. All trials were repeated 3 times.

**Table 1 foods-12-00285-t001:** Km and Vmax of HRP and nanomaterials.

Catalyst	Km/mM		Vmax/10^−8^ M s^−1^		References
	TMB	H_2_O_2_	TMB	H_2_O_2_	
HRP	0.434	3.7	10	8.7	[28]
Fe_3_O_4_	0.098	154	3.44	9.78	[29]
Fe_3_O_4_ NPs	0.313	0.013	13.35	2.95	[30]
Fe-COF	0.02	0.143	3.83	4.74	[31]
VcFe_3_O_4_ NPs	0.067	2.96	1.93	2.05	[24]
Fe_3_O_4_-MnO_2_	0.101	0.041	0.57	2.94	[32]
Fe_3_O_4_@C	0.27	0.035	12	3.34	[28]
MMIP NPs	0.088	0.16	3.44	6.25	This work

**Table 2 foods-12-00285-t002:** Recovery rate experiment results for milk spiked with NOR (*n* = 3).

Milk Sample	Spiked (ng/mL)	Detected (ng/mL)	Recovery (%)	RSD (%)
1	10.00	8.98	89.83	4.32
	50.00	42.74	85.48	3.47
	100.00	78.29	78.29	9.35
2	10.00	9.09	90.88	9.46
	50.00	44.53	89.07	6.36
	100.00	82.83	82.83	3.89
3	10.00	9.58	95.81	9.88
	50.00	41.15	82.31	6.64
	100.00	83.71	83.71	4.22

**Table 3 foods-12-00285-t003:** Comparison of detection methods for NOR.

Method	Sample	Liner Ranges (ng/mL)	LOD (ng/mL)	Recovery (%)	Reference
HPLC analysis	Milk	1 × 10^3^–30 × 10^3^	260.0	99.50–101.20	[36]
HPLC with fluorescence	chicken tissues	10–1 × 10^3^	2.5	79.20–93.40	[37]
ELISA method	water	0.1–10	0.016	74-105	[38]
Fluorescence analysis	Fish, milk	1–90	0.80	90.92–111.53	[39]
Fluorescence analysis based on immunoassay	Milk	13–3986	0.0034	86.44–101.30	[8]
Fluorescence analysis based on time-resolved methodology	Milk	0.5–1000	0.14	91.90–110.50	[40]
Raman spectroscopy	Milk	7.98–159.67	1.701	101.29–104.00	[41]
Colorimetric chemosensor based on Fe_3_O_4_ MMIP NPs	Milk	10–300	8.90	75.97–95.81	This work

## Data Availability

All related data and methods are presented in this paper and the Appendix A.

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
