# Peer review of "Colorimetric Chemosensor Based on Fe3O4 Magnetic Molecularly Imprinted Nanoparticles for Highly Selective and Sensitive Detection of Norfloxacin in Milk"

_foods, 2023, doi:10.3390/foods12020285_

Round 1

Reviewer 1 Report

The manuscript is concerned with the development of a MIP colorimetric sensor for the selective determination of a fluoroquinolone antibiotic in solution. The method was applied to real-world samples (milk), which confirms it viability.

Major concerns.

1.       line 234 – NOR  could be adsorbed by  Fe3O4 MMIPs NPs, thus occupying more cavities on the surface of Fe3O4 MMIPs NPs, preventing the contact between H2O2  and Fe3O4  MMIPs NPs, and ultimately inhibiting the peroxidase-like catalytic activity of Fe3O4  MMIPs NPs. – This explanation of the inhibiting action of the analyte seems doubtful, since NOR amount is unsufficient to competely fill the magnetite surface. Alternative explanations should be sought. One of those is a direct inhibiting effect of NOR on the TMB – H2O2 reaction. To test this hypothesis, the authors should show the absorbance spectra of the system TMB – H2O2 with and without NOR in the absence of any MIP/NIP particles.

2.       Paragraph 3.5 has nothing to do with the actual mechanism of the reaction, as it is a piece of formal kinetics. To my view, it does not add to the understanding of the real chemistry of this reaction and can be safely omitted from the paper.

3.       Fluoroquinolones at this concentration level can be determined by their intrinsic fluorescence; this possibility should be discussed when comparing the methods.

Minor remarks.

4.       line 42 – single catalytic active site.  – Make sense?

5.       line 136 – here and below, µL should be instead of uL.

6.       line 197 – H2O2 does not dissociate at these pH values, this explanation should be removed.

7.       Fig. 2 caption: (A) the pH of the experiment with the buffer type should be indicated, as well as H2O2 concentration and reaction time; (B) the TMB and H2O2 concentration and reaction time for this experiment should be shown; (c) the concentrations and reaction time should be shown; (D) all the concentrations should be shown.

8.       line 220 - Considering the conservation and environmental protection - Make sense (“conservation”)?

9.       line 224 - When the reaction time was 4 min, the solution color changed significantly in a short time, which was easy to be recognized by the naked eye” probably, should sound like “When the reaction time was 4 min, the solution color changed sufficiently to be recognized by the naked eye”.

10.   line 240 and abstract – the limit of detection is given with 1 significant digit (9 ng/mL).

11.   English should be polished.

Author Response

Dear professor,

 Thank you for your letter and the reviewers’ comments concerning our manuscript entitled “Colorimetric Chemosensor based on Fe3O4 magnetic molecularly imprinted nanoparticles for highly selective and sensitive detection of Norfloxacin in milk” (foods-2103553). Those comments are valuable and very helpful. We have studied reviewers’ comments carefully and tried our best to revise our manuscript according to the comments. Based on the instructions provided in your letter, we uploaded the file of the revised manuscript and a revised manuscript with the correction sections red marked. The responses to the reviewer's comments are marked in bold and presented following. We deeply appreciate your consideration of our manuscript, and we hope that the manuscript can meet the requirements for publication in Foods.

Comment 1. line 234 – NOR could be adsorbed by Fe3O4 MMIPs NPs, thus occupying more cavities on the surface of Fe3O4 MMIPs NPs, preventing the contact between H2O2 and Fe3O4 MMIPs NPs, and ultimately inhibiting the peroxidase-like catalytic activity of Fe3O4 MMIPs NPs. – This explanation of the inhibiting action of the analyte seems doubtful, since NOR amount is unsufficient to competely fill the magnetite surface. Alternative explanations should be sought. One of those is a direct inhibiting effect of NOR on the TMB – H2O2 reaction. To test this hypothesis, the authors should show the absorbance spectra of the system TMB – H2O2 with and without NOR in the absence of any MIP/NIP particles.

Thank you for your suggestion. We apologize for insufficient explanation which led to your misunderstanding. We have reorganized this part in line 224-230 of page 5-6. We reserved the original explanation instead of the direct inhibiting effect of NOR on the TMB – H2O2 reaction for the following reasons.

First, the surface of the Fe3O4 MMIPs NPs was modified by -NH2 and surface modification layer, which could partly decrease the peroxidase-like catalytic activity of Fe3O4 MMIPs NPs . Second, the principle of this detection method is that: We designed cavities consistent with the molecular structure of NOR on the surface of Fe3O4 MMIPs NPs to make sure that when NOR and Fe3O4 MMIPs NPs appeared at the same time NOR occupied part of the cavities on the surface of Fe3O4 MMIPs NPs and prevented the contact between H2O2 and Fe3O4 MMIPs NPs, and ultimately inhibited catalyze the oxidation of colorless TMB into a blue oxTMB.

Comment 2. Paragraph 3.5 has nothing to do with the actual mechanism of the reaction, as it is a piece of formal kinetics. To my view, it does not add to the understanding of the real chemistry of this reaction and can be safely omitted from the paper.

 Thank you for your suggestion. After careful consideration, we think that paragraph 3.5 can help to comfirm the peroxidase-like catalytic activity of Fe3O4 MMIPs NPs and it is the basis for our detection method. Therefore, we think that it’s better to show this part as 3.1 in page 4.

Comment 3. Fluoroquinolones at this concentration level can be determined by their intrinsic fluorescence; this possibility should be discussed when comparing the methods.

Thank you for your suggestion. We have compared the methods in paragraph 3.6 in line 331-340 of page 9.

Comment 4. line 42 – single catalytic active site.  – Make sense?

Thank you for your suggestion. We have deleted the “single catalytic active site” in line 44-45 of page 1.

Comment 5. line 136 – here and below, µL should be instead of uL.

Thank you for your suggestion. We have changed uL to µL in line 125 - 145 of page 3.

Comment 6. line 197 – H2O2 does not dissociate at these pH values, this explanation should be removed.

Thank you for your suggestion. We have deleted the explanation in line 230-231 of page 5.

Comment 7. Fig. 2 caption: (A) the pH of the experiment with the buffer type should be indicated, as well as H2O2 concentration and reaction time; (B) the TMB and H2O2 concentration and reaction time for this experiment should be shown; (c) the concentrations and reaction time should be shown; (D) all the concentrations should be shown.

Thank you for your suggestion. We have shown the information in Fig. 2 caption in line 216-222 of page 5. Figure 2. The influence of different factors on the performance of Fe3O4 MMIPs NPs. (A) Buffer type, 0.2 M buffer pH=3.5, 250 uL 0.06 M H2O2, 50 uL 0.008 M TMB, reaction time 4 min; (B) pH, 0.2 M HAc-NaAc, 250 uL 0.06 M H2O2, 50 uL 0.008 M TMB, reaction time 4 min; (C) TMB/H2O2 volume ratio, 0.2 M HAc-NaAc pH=3.5, 0.06 M H2O2, 0.008 M TMB, reaction time 4 min; (D) H2O2 concentration, 0.2 M HAc-NaAc pH=3.5, 250 uL H2O2, 50 uL 0.008 M TMB, reaction time 4 min; (E) Reaction time, 0.2 M HAc-NaAc pH=3.5, 250 uL 0.06 M H2O2, 50 uL 0.008 M TMB.

Comment 8. line 220 - Considering the conservation and environmental protection - Make sense (“conservation”)?

Thank you for your suggestion. We have deleted the sentence “Considering the conservation and environmental protection” in line 240 of page 5.

Comment 9. line 224 - When the reaction time was 4 min, the solution color changed significantly in a short time, which was easy to be recognized by the naked eye” probably, should sound like “When the reaction time was 4 min, the solution color changed sufficiently to be recognized by the naked eye”.

Thank you for your suggestion. We have changed the sentenceWhen the reaction time was 4 min, the solution color changed significantly in a short time, which was easy to be recognized by the naked eye” to “When the reaction time was 4 min, the solution color changed sufficiently to be recognized by the naked eye” in line 243-246 of page 5-6.

Comment 10. line 240 and abstract – the limit of detection is given with 1 significant digit (9 ng/mL).

Thank you for your suggestion. After careful consideration, we kept 3 significant digits in the limit of detection and we still used 8.90 ng/mL.

Question 11. English should be polished.

Thank you for your suggestion. The language presentation was improved with assistance from a native English speaker with appropriate research background. We hope the readability is better now.

Yours Sincerely

Xia Liu

College of Food Science and Technology, Hunan Provincial Key Laboratory of Food Science and Biotechnology, Hunan Agricultural University

Email address: liuxiaspr@aliyun.com

Reviewer 2 Report

The readability of this MS is good. The development of this novel tool has been well described by adding all useful info for replication studies. The applicability for milk samples analysis was reported and the references section is useful for the reader.

Major revision needed.

-         The most important point that needs improvement is the novelty in the field and the real applications. In this regard, the authors should reply to this question: Why a laboratory manager (or a scientist, generally speaking) should choose this approach for detecting this residue in milk samples, in the place of other, well-established, approaches. This answer should be properly addressed in a dedicated Discussion section.

-          Another point to improve is the food safety topic. The authors should focus on the food safety aspects, particularly antimicrobial resistance, baby food and legislative approaches adopted in USA, China and Europe.

-          Statistical analysis is missing. The authors should add the statistical analysis and related comments and discussion. This aspect also relies with method validation. Please add all useful info about method validation (accuracy, selectivity, robustness, etc.).

Other minor points:

Line 16: 8.9 ng/mL

Please add a reference at line 28.

Please improve readability, check keying errors and standardize the font at lines 37, 46, 74, 244, 260, 269-273.

Line 78: what is the meaning of “inexpensive”? Please improve.

Lines 92-93: please specify the meaning of all acronyms the first time cited.

Page 3: please add the uncertainty value (if available) for each temperature indicated. Please add spacing: 25 °C at lines 113, 122 and 139.

Line 108: please specify how the pH value was corrected.

Line 110: Five hundred mg…..

Line 114: please improve reproducibility adding more info about vacuum drying step.

Line 126: One hundred µL (please always use µ in the place of u).

Par. 2.3: please add more info about the spectrophotometer.

Line 133 and 259: please standardize the font (comma).

Line 146: One mL of milk…

Line 148: please specify temperature and frequency of ultrasound step.

Line 149: why these levels? Please explain.

Line 161: what about statistics? Please improve.

Par. 3.1: please specify how many repetitions were made for these trials.

Please avoid abbreviations at lines 181 and 183.

Lines 214-226: what about the possibility of increasing sensitivity by increasing H2O2 concentration and/or reaction time?

Lines 237-238: how many repetitions were made? Please specify.

Line 240: Is this a LOD or LOQ? Please specify and describe the approach used for its determination.

Lines 254-256: why these tests were not continued over than 6 re-use?

Lines 264-268: Please improve. Please add more info about samples and add data demonstrating the consistency of results with ELISA kit. This last point is very significant and it deserves specific comments.

Table 1: please use subscript in the caption, where needed.

Par. 3.5: Please use KM in the place of Km

Figures: please add spacing in some axis labels, i.e., Absorbance (a.u.), Wavelenght (nm).

Please check the following references for comparison:

Qiu et al (2022) Sensitive determination of Norfloxacin in milk based on β-cyclodextrin functionalized silver nanoparticles SERS substrate. Spectrochimica Acta Part A: Molecular and Biomolecular Spectroscopy, 276, https://doi.org/10.1016/j.saa.2022.121212.

Rodríguez‐Díaz et al (2004) Sensitive Determination of Fluoroquinolone Antibiotics in Milk Samples Using Time‐Resolved Methodology. Analytical Letters, 37:6, 1163-1175, DOI: 10.1081/AL-120034061 .

Author Response

Dear professor,

 Thank you for your letter and the reviewers’ comments concerning our manuscript entitled “Colorimetric Chemosensor based on Fe3O4 magnetic molecularly imprinted nanoparticles for highly selective and sensitive detection of Norfloxacin in milk” (foods-2103553). Those comments are valuable and very helpful. We have studied reviewers’ comments carefully and tried our best to revise our manuscript according to the comments. Based on the instructions provided in your letter, we uploaded the file of the revised manuscript and a revised manuscript with the correction sections red marked. The responses to the reviewer's comments are marked in bold and presented following. We deeply appreciate your consideration of our manuscript, and we hope that the manuscript can meet the requirements for publication in Foods.

Comment 1. The most important point that needs improvement is the novelty in the field and the real applications. In this regard, the authors should reply to this question: Why a laboratory manager (or a scientist, generally speaking) should choose this approach for detecting this residue in milk samples, in the place of other, well-established, approaches. This answer should be properly addressed in a dedicated Discussion section.

Thank you for your suggestion. We have compared the method in this work with other reported methods and further discussed the advantages of our work in line 341-347 of page 9.

Comment 2. Another point to improve is the food safety topic. The authors should focus on the food safety aspects, particularly antimicrobial resistance, baby food and legislative approaches adopted in USA, China and Europe.

Thank you for your suggestion. We have described the food safety topic in introduction in line 28-30 of page 1. We found that EU regulated the permitted maximum limit of NOR in animal products (1 ppm) and China issued regulations on the suspension of production, operation and usage of NOR. This information has been added in line 30-32 of page 1.

Even though we didn’t find regulation about NOR from USA, we found that USA issued regulations on the suspension of production, operation and usage of enoxacin, another kind of fluoroquinolones antibiotics.

Comment 3. Statistical analysis is missing. The authors should add the statistical analysis and related comments and discussion. This aspect also relies with method validation. Please add all useful info about method validation (accuracy, selectivity, robustness, etc.).

  Thank you for your suggestion. We have added statistical analysis and related comments and discussion, for example Figure 3 C and Figure 3 D.

  The information about method validation have been described in the paper. For example, Table 1 Recovery rate experiment results for milk spiked with NOR showed the accuracy, Figure 3 C showed the selectivity.

Comment 4. Line 16: 8.9 ng/mL

Thank you for your suggestion. After careful consideration, we kept 3 significant digits in the limit of detection and we still used 8.90 ng/mL.

Comment 5. Please add a reference at line 28.

Thank you for your suggestion. We have added a reference in line 33.

Comment 6. Please improve readability, check keying errors and standardize the font at lines 37, 46, 74, 244, 260, 269-273.

Thank you for your suggestion. We have checked and corrected the errors and standardize the font at lines 37, 46, 74, 244, 260, 269-273.

Comment 7. Line 78: what is the meaning of “inexpensive”? Please improve.

Thank you for your suggestion. We have changed “inexpensive” to “low-cost” in line 77 of page 2.

Comment 8. Lines 92-93: please specify the meaning of all acronyms the first time cited.

Thank you for your suggestion. We have specified the meaning of all acronyms the first time cited.

Comment 9. Page 3: please add the uncertainty value (if available) for each temperature indicated.

Thank you for your suggestion. We have checked page 3 and there is no uncertainty value for each temperature indicated.

Comment 10. Please add spacing: 25 °C at lines 113, 122 and 139.

Thank you for your suggestion. We have added spacing: 25 °C all over the manuscript.

Comment 11. Line 108: please specify how the pH value was corrected.

Thank you for your suggestion. 100 mM Tris water solution was first prepared and the pH was corrected to 8.5 use HCl. Then the Tris-HCl was mixed with ethanol (1:1, v/v).

Comment 12. Line 110: Five hundred mg…..

Thank you for your suggestion. We have changed 500 mg to five hundred mg.

Comment 13. Line 114: please improve reproducibility adding more info about vacuum drying step.

Thank you for your suggestion. The vacuum drying step was carried out under 60 ℃ for 24 h. We have added the information in line 118 of page 3.

Comment 14. Line 126: One hundred µL (please always use µ in the place of u).

Thank you for your suggestion. We have changed uL to µL in line 125 - 145 of page 3.

Comment 15. Par. 2.3: please add more info about the spectrophotometer.

Thank you for your suggestion. We used Multi-Mode microplate spectrophotometer (Multiskan GO 1510, Thermo Fisher Scientific, USA) and this has been described in line 103-104 of page 2.

Comment 16. Line 133 and 259: please standardize the font (comma).

Thank you for your suggestion. We have standardized the font of comma to half corner character in Times New Roman.

Comment 17. Line 146: One mL of milk…

Thank you for your suggestion. We have change “1 mL milk” to “One mL of milk”.

Comment 18. Line 148: please specify temperature and frequency of ultrasound step.

Thank you for your suggestion. The temperature is 25 ℃, frequency is 60hz, we have added this information in line 152 of page 3.

Comment 19. Line 149: why these levels? Please explain.

Thank you for your suggestion. These levels were chosen within the linear range (10-300 ng/mL).

Comment 20. Line 161: what about statistics? Please improve.

Thank you for your suggestion. This is the result of spectrogram and statistics analysis for spectrogram are meaningless.

Comment 21. Par. 3.1: please specify how many repetitions were made for these trials.

Thank you for your suggestion. All of these trials was repeated for three times and we have added the description in Figure 1 caption in line 204 of page 5.

Comment 22. Please avoid abbreviations at lines 181 and 183.

Thank you for your suggestion. We have avoided abbreviations all over the manuscript.

Comment 23. Lines 214-226: what about the possibility of increasing sensitivity by increasing H2O2 concentration and/or reaction time?

Thank you for your suggestion. The establishment of Linear relationship between the ΔA (652 nm) and NOR concentration is based on the optimization of H2O2 concentration and reaction time.

Comment 24. Lines 237-238: how many repetitions were made? Please specify.

Thank you for your suggestion. All of these trials was repeated for three times. We have added this description in line 296-297 of page 7.

Comment 25. Line 240: Is this a LOD or LOQ? Please specify and describe the approach used for its determination.

Thank you for your suggestion. It’s LOD. According to the rules of International Union of Pure and Applied Chemistry (IUPAC): CL=k × Sb /m (CL means the detection limit; k means theconfidence factor; m means the slope of standard curve in the low concentration range; Sb means the standard deviation of a blank. We have added this in 2.6 in line 156-159 of page 3.

Same results can be found in the following reports.

A range of 1nM-1000 nM with a detection limit of 6.18 nM. (Qingkun Kong, Yanhu Wang, Lina Zhang, Shenguang Ge, Jinghua Yu, A novel microfluidic paper-based colorimetric sensor based on molecularly imprinted polymer membranes for highly selective and sensitive detection of bisphenol A, Sensors and Actuators B: Chemical, 2017, 243: 130-136.)

The linear range of 2 × 10-5 - 6 × 10-4 mol L-1 with detection limit of 1 × 10-5 mol L-1 (Lili Guo, Huijun Zheng, Cuijie Zhang, Lingbo Qu, Lanlan Yu, A novel molecularly imprinted sensor based on PtCu bimetallic nanoparticle deposited on PSS functionalized graphene with peroxidase-like activity for selective determination of puerarin. Talanta. 2020, 210: 120621.)

Comment 26. Lines 254-256: why these tests were not continued over than 6 re-use?

Thank you for your suggestion. As we show in Figure 3 D, there is no significant difference between the 5 th and 6 th re-use. Therefore, we didn’t continue over than 6 re-use.

Comment 27. Lines 264-268: Please improve. Please add more info about samples and add data demonstrating the consistency of results with ELISA kit. This last point is very significant and it deserves specific comments.

Thank you for your suggestion. The samples were natural pure milk purchased from Inner Mongolia Yili Industrial Group Limited by Share Ltd, natural pure milk purchased from Inner Mongolia Mengniu Dairy (Group) Limited by Share Ltd., and Deluxe Milk purchased from Inner Mongolia Mengniu Dairy (Group) Limited by Share Ltd. We have added this in line 95-98 of page 2.

The results of ELISA is that: detected NOR concentration in three milk sample was -0.0906 ng/mL, -0.1660 ng/mL and -0.1376 ng/mL, respectively. We have added this in line 291 of page 7.

Comment 28. Table 1: please use subscript in the caption, where needed.

Thank you for your suggestion. We have changed the Table 1 as the suggestion.

Comment 29. Par. 3.5: Please use KM in the place of Km

Thank you for your suggestion. Km is the technical term, we don‘t think KM can be used in the place of Km.

Comment 31. Figures: please add spacing in some axis labels, i.e., Absorbance (a.u.), Wavelenght (nm).

Thank you for your suggestion. We have added spacing in axis labels in Figure 1, Figure 2, and Figure 3 A.

Comment 32. Please check the following references for comparison:

Qiu et al (2022) Sensitive determination of Norfloxacin in milk based on β-cyclodextrin functionalized silver nanoparticles SERS substrate. Spectrochimica Acta Part A: Molecular and Biomolecular Spectroscopy, 276, https://doi.org/10.1016/j.saa.2022.121212.

Rodríguez‐Díaz et al (2004) Sensitive Determination of Fluoroquinolone Antibiotics in Milk Samples Using Time‐Resolved Methodology. Analytical Letters, 37:6, 1163-1175, DOI: 10.1081/AL-120034061

Thank you for your suggestion. We have added and compared the references in line 331-340 of page 9.

Yours Sincerely

Xia Liu

College of Food Science and Technology, Hunan Provincial Key Laboratory of Food Science and Biotechnology, Hunan Agricultural University

Email address: liuxiaspr@aliyun.com

Reviewer 3 Report

Review foods-2103553

Title

Colorimetric Chemosensor based on Fe3O4 magnetic molecularly imprinted nanoparticles for highly selective and sensitive detection of Norfloxacin in milk

Authors

Maiquan Li , Lingli Luo , Jiayin Li , Yingzi Xiong , Ling Wang , Xia Liu *

The manuscript by Liu and co-workers presents the colorimetric chemosensor for detection of NOR in milk.

This is interesting paper, however  some minor corrections should be provided.

My specific comments are:

1/ I would suggest to avoid phrase “simple”, if yes, please prove that this sensor is “simple” and describe the comparison with the other sensors

2/ please explain how the detection limit was calculated, the detection range was 10-200 ng.ml, but the detection limit below this range, 8.90 ng/ml

3/ what was the lowest recovery range, 75.97 as it is written in abstract, or 78.29 presented in Table 1? Is this range too low?

4/ what was the detection limit in milk samples?

5/ is stability related to the reusability ? 6 cycles is good enough? Please compare with the similar literature examples.

Author Response

 Thank you for your letter and the reviewers’ comments concerning our manuscript entitled “Colorimetric Chemosensor based on Fe3O4 magnetic molecularly imprinted nanoparticles for highly selective and sensitive detection of Norfloxacin in milk” (foods-2103553). Those comments are valuable and very helpful. We have studied reviewers’ comments carefully and tried our best to revise our manuscript according to the comments. Based on the instructions provided in your letter, we uploaded the file of the revised manuscript and a revised manuscript with the correction sections red marked. The responses to the reviewer's comments are marked in bold and presented following. We deeply appreciate your consideration of our manuscript, and we hope that the manuscript can meet the requirements for publication in Foods.

Comment 1. I would suggest to avoid phrase “simple”, if yes, please prove that this sensor is “simple” and describe the comparison with the other sensors

Thank you for your suggestion. Compared with other reported detection method, Fe3O4 MMIPs NPs allowed the sample pretreatment perfectly combined with colorimetric detection making the whole procedure simple in this study. This discussion has been added in 341-347 of page 9.

Comment 2. please explain how the detection limit was calculated, the detection range was 10-200 ng.ml, but the detection limit below this range, 8.90 ng/ml

Thank you for your suggestion.  According to the rules of International Union of Pure and Applied Chemistry (IUPAC): CL=k × Sb /m (CL means the detection limit; k means theconfidence factor; m means the slope of standard curve in the low concentration range; Sb means the standard deviation of a blank. We have added this in 2.6 in line 156-159 of page 3.

Same results can be found in the following reports.

A range of 1nM-1000 nM with a detection limit of 6.18 nM. (Qingkun Kong, Yanhu Wang, Lina Zhang, Shenguang Ge, Jinghua Yu, A novel microfluidic paper-based colorimetric sensor based on molecularly imprinted polymer membranes for highly selective and sensitive detection of bisphenol A, Sensors and Actuators B: Chemical, 2017, 243: 130-136.)

The linear range of 2 × 10-5 - 6 × 10-4 mol L-1 with detection limit of 1 × 10-5 mol L-1 (Lili Guo, Huijun Zheng, Cuijie Zhang, Lingbo Qu, Lanlan Yu, A novel molecularly imprinted sensor based on PtCu bimetallic nanoparticle deposited on PSS functionalized graphene with peroxidase-like activity for selective determination of puerarin. Talanta. 2020, 210: 120621.)

Comment 3. what was the lowest recovery range, 75.97 as it is written in abstract, or 78.29 presented in Table 1? Is this range too low?

 Thank you for your suggestion. We apologize for the mistake we made in the abstract, the correct lowest recovery range should be 78.29. As we compared in Table 3, some reported methods was even lower than 78.29 (77.3 for ELISA method reference 36).

Comment 4. what was the detection limit in milk samples?

 Thank you for your suggestion. As described in introduction In 2015, the Ministry of Agriculture of the People's Republic of China issued regulations on the suspension of production, operation and usage of NOR in the Announcement No. 2292. The NOR was not allowed to be detected in milk.

Comment 5. Is stability related to the reusability? 6 cycles is good enough? Please compare with the similar literature examples.

Thank you for your suggestion. The stability is related to the reusability, only when the Fe3O4 MNIPs NPs is stable enough the reusability can be guarantee. We have compared with other methods in line 286-289 of page 7.

Yours Sincerely

Xia Liu

College of Food Science and Technology, Hunan Provincial Key Laboratory of Food Science and Biotechnology, Hunan Agricultural University

Email address: liuxiaspr@aliyun.com

Round 2

Reviewer 1 Report

The detection limits cannot be shown with 3 significant digits even "after a careful consideration". There is no room to "consider", they are the values known only approximately and must be changed to 1-digit numbers.

And I am quite not satisfied with the explanation given to my Comment 1: 

(Comment 1. line 234 – NOR could be adsorbed by Fe3O4 MMIPs NPs, thus occupying more cavities on the surface of Fe3O4 MMIPs NPs, preventing the contact between H2O2 and Fe3O4 MMIPs NPs, and ultimately inhibiting the peroxidase-like catalytic activity of Fe3O4 MMIPs NPs. – This explanation of the inhibiting action of the analyte seems doubtful, since NOR amount is unsufficient to competely fill the magnetite surface. Alternative explanations should be sought. One of those is a direct inhibiting effect of NOR on the TMB – H2O2 reaction. To test this hypothesis, the authors should show the absorbance spectra of the system TMB – H2O2 with and without NOR in the absence of any MIP/NIP particles.)

I am insisting on changing this explanation and measuring the spectra I am talking about.

Author Response

 Thank you for your comments concerning our manuscript entitled “Colorimetric Chemosensor based on Fe3O4 magnetic molecularly imprinted nanoparticles for highly selective and sensitive detection of Norfloxacin in milk” (foods-2103553). Those comments are valuable and very helpful. We have studied reviewers’ comments carefully and tried our best to revise our manuscript according to the comments. Based on the instructions provided in your letter, we uploaded the file of the revised manuscript and a revised manuscript with the correction sections red marked. The responses to the reviewer's comments are marked in bold and presented following. We deeply appreciate your consideration of our manuscript, and we hope that the manuscript can meet the requirements for publication in Foods.

Reply to Reviewer: 1

Comment 1. The detection limits cannot be shown with 3 significant digits even "after a careful consideration". There is no room to "consider", they are the values known only approximately and must be changed to 1-digit numbers.

 Thank you for your suggestion. We have changed 8.90 ng/mL to 9 ng/mL in line 16 of page 1 and in line 258 of page 7.

Comment 2.

line 234 – NOR could be adsorbed by Fe3O4 MMIPs NPs, thus occupying more cavities on the surface of Fe3O4 MMIPs NPs, preventing the contact between H2O2 and Fe3O4 MMIPs NPs, and ultimately inhibiting the peroxidase-like catalytic activity of Fe3O4 MMIPs NPs. – This explanation of the inhibiting action of the analyte seems doubtful, since NOR amount is unsufficient to competely fill the magnetite surface. Alternative explanations should be sought. One of those is a direct inhibiting effect of NOR on the TMB – H2O2 reaction. To test this hypothesis, the authors should show the absorbance spectra of the system TMB – H2O2 with and without NOR in the absence of any MIP/NIP particles.

Thank you for your suggestion. We have changed the explanations in line 246-256 of page 6 and shown the absorbance spectra of the system TMB – H2O2 with and without NOR in the absence of any MIP/NIP particles in the Figure 1B. We hope it’s clear enough now.

Yours Sincerely

Xia Liu

College of Food Science and Technology, Hunan Provincial Key Laboratory of Food Science and Biotechnology, Hunan Agricultural University

Email address: liuxiaspr@aliyun.com

Reviewer 2 Report

The authors made most important revisions requested.

Minor revision needed.

Line 344: needs

Table 3: Linear range. Please also use the same number of significant figures in recovery column.

Line 32: please specify the matrix

Please correct some keying errors in Fig.3 caption (comma and 25 °C)

Line 16: the second decimal place makes no sense. Please remove it.

Line 80: was examined

Fig. 1-3 captions: please use: All trials repeated 3 times

Line 294: please replace “exist” by more proper term.

The font used for “° C” seems not standardized throughout the text. Please verify.

Line 141: One hundred….

Lines 146-149: please standardize the font use for comma.

Line 151: Ten µL….

Line 163: 60 Hz

Please check parenthesis at par. 2.6.

Please avoid abbreviations at lines 219 and 221.

Please check some keying errors at lines 324.

Author Response

Dear professor,

 Thank you for your comments concerning our manuscript entitled “Colorimetric Chemosensor based on Fe3O4 magnetic molecularly imprinted nanoparticles for highly selective and sensitive detection of Norfloxacin in milk” (foods-2103553). Those comments are valuable and very helpful. We have studied reviewers’ comments carefully and tried our best to revise our manuscript according to the comments. Based on the instructions provided in your letter, we uploaded the file of the revised manuscript and a revised manuscript with the correction sections red marked. The responses to the reviewer's comments are marked in bold and presented following. We deeply appreciate your consideration of our manuscript, and we hope that the manuscript can meet the requirements for publication in Foods.

Reply to Reviewer: 2

Comment 1. Line 344: needs

Thank you for your suggestion. We have changed “need” to “needs” in line 301-302 of page 8.

Comment 2. Linear range. Please also use the same number of significant figures in recovery column.

Thank you for your suggestion. We have used the same number of significant figures in recovery column.

Comment 3. Line 32: please specify the matrix

   Thank you for your suggestion. The matrix are animal products, we have added this information in line 30-32 of page 1.

Comment 4. Please correct some keying errors in Fig.3 caption (comma and 25 °C)

Thank you for your suggestion. We have corrected the errors in Fig.3 caption in line 272-274 of page 7.

Comment 5. Line 16: the second decimal place makes no sense. Please remove it.

Thank you for your suggestion. As reviewer 1 suggested, we have changed 8.90 ng/mL to 9 ng/mL in line 16 of page 1 and in line 258 of page 7.

Comment 6. Line 80: was examined

Thank you for your suggestion. The “were examined” has changed to “was examined”.

Comment 7. Fig. 1-3 captions: please use: All trials repeated 3 times

Thank you for your suggestion. Fig. 1-3 caption “All of these trials was repeated for three times.” has been changed to “All trials repeated 3 times.”

Comment 8. Line 294: please replace “exist” by more proper term.

Thank you for your suggestion. we have changed the sentence “Figure 3 C showed ΔA of Fe3O4 MMIPs NPs-H2O2-TMB reaction system and Fe3O4 MNIPs NPs-H2O2-TMB reaction system with the exist of NOR, CIP, ENR, DAN, SD and TC” to “Figure 3 C showed ΔA of Fe3O4 MMIPs NPs-H2O2-TMB reaction system with and without NOR, CIP, ENR, DAN, SD and TC” in line 254-255 of page 7.

Comment 9. The font used for “° C” seems not standardized throughout the text. Please verify.

Thank you for your suggestion. We have standardized “° C” in line 100-138 of page 2-3.

Comment 10. Line 141: One hundred….

Thank you for your suggestion. We have changed “100” to “One hundred” in line 117 of page 3.

Comment 11. Lines 146-149: please standardize the font use for comma.

Thank you for your suggestion. We have standardized the font use for comma in line 122-125 of page 3 and in line 272 of page 7.

Comment 12. Line 151: Ten µL….

Thank you for your suggestion. We have changed 10 µL to Ten µL in line 126 of page 3.

Comment 13. Line 163: 60 Hz

Thank you for your suggestion. We have changed 60 hz to 60 Hz in line 138 of page 3.

Comment 14. Please check parenthesis at par. 2.6.

Thank you for your suggestion. We have checked parenthesis at par. 2.6 and added “)” in line 145 of page 3.

Comment 15. Please avoid abbreviations at lines 219 and 221.

Thank you for your suggestion. We have deleted all of the abbreviations in line 219 and 221.

Comment 16. Please check some keying errors at lines 324.

Thank you for your suggestion. we have corrected the error in line 332.

Yours Sincerely

Xia Liu

College of Food Science and Technology, Hunan Provincial Key Laboratory of Food Science and Biotechnology, Hunan Agricultural University

Email address: liuxiaspr@aliyun.com
